# On the Limitation of Local Intrinsic Dimensionality for Characterizing the Subspaces of Adversarial Examples

**Pei-Hsuan Lu**
National Chung Hsing University
Taichung, Taiwan

**Pin-Yu Chen**
IBM Research
NY, USA

**Chia-Mu Yu**
National Chung Hsing University
Taichung, Taiwan

## Abstract

Understanding and characterizing the subspaces of adversarial examples aid in studying the robustness of deep neural networks (DNNs) to adversarial perturbations. Very recently, Ma et al. (2018) proposed to use local intrinsic dimensionality (LID) in layer-wise hidden representations of DNNs to study adversarial subspaces. It was demonstrated that LID can be used to characterize the adversarial subspaces associated with different attack methods, e.g., the Carlini and Wagner's (C&W) attack and the fast gradient sign attack.

In this paper, we use MNIST and CIFAR-10 to conduct two new sets of experiments that are absent in existing LID analysis and report the limitation of LID in characterizing the corresponding adversarial subspaces, which are (i) oblivious attacks and LID analysis using adversarial examples with different confidence levels; and (ii) black-box transfer attacks. For (i), we find that the performance of LID is very sensitive to the confidence parameter deployed by an attack, and the LID learned from ensembles of adversarial examples with varying confidence levels surprisingly gives poor performance. For (ii), we find that when adversarial examples are crafted from another DNN model, LID is ineffective in characterizing their adversarial subspaces. These two findings together suggest the limited capability of LID in characterizing the subspaces of adversarial examples.

## 1 Introduction and Background

In recent years, many studies have shown that well-trained DNNs are quite vulnerable to adversarial examples Szegedy et al. (2013); Goodfellow et al. (2015). To fundamentally understand the origins of adversarial examples and to enhance the robustness of DNNs to adversarial perturbations, many efforts have been put into differentiating adversarial and normal examples based on characterizing the subspaces they reside in. For example, MagNet Meng & Chen (2017) uses an adversary detector and a data reformer learned from the data manifold of natural examples for defending adversarial examples, and it has demonstrated robust defense performance against the powerful C&W attack under different confidence levels in the *oblivious* attack setting, where the adversarial examples are generated from an undefended DNN and are oblivious to the deployed defenses on the same DNN.

More recently, Ma et al. (2018) proposed to characterize adversarial subspaces by using local intrinsic dimensionality (LID) Karger & Ruhl (2002); Houle et al. (2012). Given a reference sample $x$ generated by some distribution $\mathcal{P}$, the maximum likelihood estimator (MLE) of LID is defined as

$$\widehat{LID}(x) = -\left( \frac{1}{k} \sum_{i=1}^{k} \log \frac{r_i(x)}{r_k(x)} \right)^{-1},$$ (1)

where $r_i(x)$ denotes the distance between $x$ and its $i$-th nearest neighbor within a sample of points drawn from $\mathcal{P}$, and $r_k(x)$ denotes the maximum distance among $k$ nearest neighbors. In Ma et al. (2018), the MLE estimate of LID is separately applied to the layer-wise hidden representations of DNNs (or simply the last (logits) layer) of adversarial and normal examples. When characterizing adversarial subspaces, additional random perturbations are also included in the LID detector to dif-

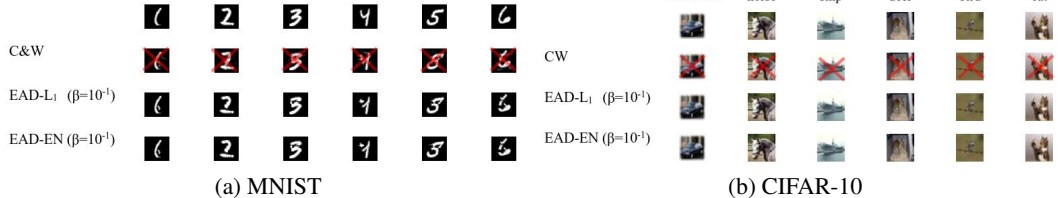

|            | (a) MNIST | (b) CIFAR-10 |

Figure 1: Visual illustration of adversarial examples crafted by different attack methods at confidence $\kappa = 20$ in the transfer attack setting. Unsuccessful attacks detected by the respective LID-detectors are marked by red cross sign. EAD Chen et al. (2017) can yield highly transferable and visually similar adversarial examples, whereas the C&W attack Carlini & Wagner (2017) is less likely to bypass the detector. The quantitative evaluation is presented in Table 1.

ferentiate random and adversarial perturbations. Although LID is proposed for subspace analysis rather than defense, it exhibits strong discrimination power against five state-of-the-art attacks.

In this paper, in addition to the C&W attack, we also consider an untested attack called elastic-net attacks to DNNs (EAD), an $L_1$-distortion-based method proposed in Chen et al. (2017). EAD is a generalized attack that includes the C&W attack as a special case when the $L_1$ penalty coefficient $\beta = 0$. In principle, larger $\beta$ can generate more effective and transferable adversarial examples Chen et al. (2017); Sharma & Chen (2017). We conduct two new sets of experiments that are absent in Ma et al. (2018) and report the limitation of LID in characterizing the respective adversarial subspaces:

**1. Oblivious attack with varying confidence.** In Ma et al. (2018), for the C&W attack the adversarial examples used to train the LID detector are those with the confidence parameter $\kappa$ set to be 0, i.e., low-confidence adversarial examples. Although the recent work in Athalye et al. (2018) has demonstrated that high-confidence (larger $\kappa$) adversarial examples can bypass the same detector, it is not clear whether the detector can be strengthened if trained on high-confidence adversarial examples. Here we find that even the LID detector is trained using the *same* attack and confidence, at low confidences its performance, in terms of AUC score Ma et al. (2018) and attack detection rate, is only significant against C&W attack and is much less effective against EAD. Perhaps more surprisingly, we trained the LID detector using ensembles of adversarial examples with varying confidence levels and find that the resulting detector gives poor detection performance for all confidence levels.

**2. Transfer attack.** Adversarial examples are known to be transferable from one DNN model to another Papernot et al. (2016); Liu et al. (2016). We find that when adversarial examples are crafted from another DNN model (with similar classification accuracy), LID trained using the *same* attack and confidence on the target model is still ineffective in characterizing adversarial subspaces at high confidences, especially for $L_1$-norm-based adversarial examples crafted by EAD. Figure 1 shows some visual comparison of adversarial examples crafted by C&W attack and EAD in this setting.

Our findings suggest the limited capability of LID in characterizing the subspaces of adversarial examples. In particular, without knowing the attack parameters, in the oblivious attack setting the performance of LID detector can be severely degraded. In the transfer attack setting, even knowing the attack parameters, LID appears to overlook the subspaces of transferable adversarial examples.

## 2 EXPERIMENTS

### 2.1 EXPERIMENT SETUP AND PARAMETER SETTING

We use MNIST and CIFAR-10 to perform untargeted adversarial attacks in the aforementioned experiments. 1000 correctly classified images are randomly selected from the test sets as attack targets. In the oblivious attack setting, we use the DNN models[1] provided by Ma et al. (2018). In the transfer attack setting, we use different DNN models[2] in Carlini & Wagner (2017) to craft adversarial examples. For EAD attack, we use the default implementation[3] and set the $L_1$ regularization coefficient

---

[1] https://github.com/xingjunm/lid_adversarial_subspace_detection

[2] https://github.com/carlini/nn_robust_attacks.

[3] https://github.com/ysharma1126/EAD_Attack

Table 1: Transfer attacks on LID-based subspace analysis trained with the *same* attack and confidence on the target model. Detection rate means the attacks detected by the respective detectors.

| Attack method | $\kappa$ | MNIST | | | CIFAR-10 | | |
|---|---|---|---|---|---|---|---|
| | | detection rate (%) | classification rate (%) (post detection) | classification rate w/o detection (%) | detection rate (%) | classification rate (%) (post detection) | classification rate w/o detection (%) |
| C&W($L_2$) | 0 | 25.90 | 73.48 | 98.89 | 1.01 | 95.28 | 96.30 |
| | 10 | 39.41 | 54.93 | 90.42 | 9.68 | 87.26 | 94.52 |
| | 20 | 34.67 | 36.79 | 57.96 | 16.05 | 79.49 | 90.95 |
| | 30 | 38.10 | 11.08 | 23.38 | 17.45 | 75.92 | 88.15 |
| | 40 | 36.29 | 2.31 | 5.24 | 17.70 | 72.73 | 84.45 |
| EAD (EN rule) | 0 | 38.20 | 59.37 | 94.65 | 14.52 | 81.14 | 93.63 |
| | 10 | 47.47 | 33.56 | 64.51 | 15.41 | 78.21 | 88.28 |
| | 20 | 36.29 | 15.32 | 28.72 | 21.78 | 68.66 | 81.78 |
| | 30 | 29.43 | 4.73 | 8.16 | 21.14 | 65.09 | 77.45 |
| | 40 | 31.25 | 0.80 | 1.81 | 23.43 | 57.19 | 69.29 |
| EAD ($L_1$ rule) | 0 | 38.70 | 58.66 | 94.65 | 6.24 | 88.91 | 93.63 |
| | 10 | 47.68 | 34.37 | 64.91 | 16.81 | 77.83 | 88.02 |
| | 20 | 35.68 | 15.32 | 27.82 | 19.10 | 71.46 | 83.69 |
| | 30 | 30.04 | 5.44 | 9.27 | 22.03 | 62.42 | 75.66 |
| | 40 | 31.35 | 0.90 | 1.91 | 22.42 | 58.47 | 70.70 |

$\beta = 0.1$. We also report the EAD attack results using different decision rules (elastic-net (EN) or $L_1$ distortion) in selecting adversarial examples. For C&W attack, we use its default setting[2]. The confidence level $\kappa$ of C&W attack an EAD is picked from the range of [0, 40]. For training LID-based detectors, we use the default setting[1] in Ma et al. (2018) using all DNN layers and set the number of neighbors $k = 20$ and use a minibatch of size 100. Note that in the following experiments, unless specified, the LID detectors are trained with the *same* attacks and the *same* confidence levels, which already weakens the attack capability in the best plausible way. All experiments are conducted using an Intel Xeon E5-2620v4 CPU, 125 GB RAM and a NVIDIA TITAN Xp GPU with 12 GB RAM.

## 2.2 EVALUATION OF OBLIVIOUS ATTACKS ON LID-BASED SUBSPACE ANALYSIS

We report that even if the LID detectors are trained with the *same* attack and confidence $\kappa$, the respective detectors, in most cases, can only detect less than $50\%$ of the adversarial examples on CIFAR-10 and around $35 - 60\%$ on MNIST. At a low confidence ($\kappa = 0$), on CIFAR-10 the LID detector can detect roughly 89% of adversarial examples from C&W attack, but can only detect 42% of those from EAD with the EN decision rule. Both the area under curve (AUC) score used in Ma et al. (2018) and the detection rate tend to drop significantly as the confidence increases, especially when $\kappa = 20$ and 30. The complete results are presented in Table 2 of the supplementary material.

We proceed to check if a LID detector can be made more robust when trained with the *same* attack and *ensembles* of confidence $\kappa = \{0, 10, 20, 30, 40\}$ when crafting adversarial examples. Interestingly and perhaps surprisingly, when compared to LID detector using single $\kappa$, we find that training with more adversarial examples (5 times more in this case) of different confidences actually ruins its performance. The detection rates and the AUC scores on these two datasets are consistently below 45% and 85% for all attacks, respectively, and the AUC score of MNIST is much lower than that of CIFAR-10. The complete results are presented in Table 3 of the supplementary material.

## 2.3 EVALUATION OF TRANSFER ATTACKS ON LID-BASED SUBSPACE ANALYSIS

Table 1 shows that when LID is used in adversary detection, its detection rate can be further decreased by transfer attacks, even if the detectors are trained with the same attack and confidence on the target model. On MNIST, when $\kappa = 40$ the detection rate of EAD is around 31% and its post-detection classification rate of undetected attacks is less than 1%. On CIFAR-10, EAD exhibits better attack transferability than C&W attack in terms of the post-detection classification rate.

## 3 CONCLUSION

We summarize the main results of this paper as follows:
1. Even at low confidences and given complete attack parameters for analysis, LID is still ineffective in characterizing the subspaces of $L_1$-distortion-based adversarial examples crafted by EAD.
2. LID-based subspace analysis using ensembled confidence training leads to worse performance.
3. LID has limited capability in characterizing the subspaces of transferable adversarial examples.

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

SUPPLEMENTARY MATERIAL

Table 2: Oblivious attacks on LID-based subspace analysis trained with the *same* attack and confidence $\kappa$ on MNIST and CIFAR-10. AUC and detection rate mean the area under curve (%) and the fraction of attacks (%) detected by the respective detectors.

| Attack method | $\kappa$ | MNIST | | CIFAR-10 | |
|---|---|---|---|---|---|
| | | AUC score | detection rate | AUC score | detection rate |
| C&W($L_2$) | 0 | 89.19 | 58.56 | 99.05 | 89.17 |
| | 10 | 85.99 | 53.83 | 93.87 | 65.35 |
| | 20 | 78.61 | 36.79 | 83.24 | 43.18 |
| | 30 | 77.75 | 31.45 | 82.87 | 43.31 |
| | 40 | 81.94 | 48.89 | 82.87 | 40.38 |
| EAD (EN rule) | 0 | 87.85 | 60.38 | 82.22 | 42.92 |
| | 10 | 84.48 | 53.52 | 92.31 | 60.38 |
| | 20 | 78.98 | 36.49 | 87.11 | 46.87 |
| | 30 | 78.23 | 35.28 | 82.90 | 44.33 |
| | 40 | 81.97 | 45.56 | 82.86 | 41.27 |
| EAD ($L_1$ rule) | 0 | 87.96 | 58.16 | 93.85 | 64.71 |
| | 10 | 85.18 | 51.91 | 87.15 | 49.80 |
| | 20 | 78.79 | 38.40 | 84.58 | 43.56 |
| | 30 | 78.47 | 35.98 | 81.42 | 41.52 |
| | 40 | 82.99 | 49.49 | 83.00 | 41.65 |

Table 3: Oblivious attacks on LID-based subspace analysis trained with the *same* attack and *ensembles* of confidence $\kappa = \{0, 10, 20, 30, 40\}$ on MNIST and CIFAR-10. AUC and detection rate mean the area under curve (%) and the fraction of attacks (%) detected by the respective detectors.

| Attack method | $\kappa$ | MNIST | | CIFAR-10 | |
|---|---|---|---|---|---|
| | | AUC score | detection rate | AUC score | detection rate |
| C&W($L_2$) | 0 | 82.75 | 44.75 | 84.18 | 44.58 |
| | 10 | 82.30 | 44.75 | 83.18 | 43.05 |
| | 20 | 76.19 | 31.14 | 82.24 | 42.67 |
| | 30 | 75.10 | 31.85 | 82.00 | 43.69 |
| | 40 | 77.62 | 39.21 | 82.09 | 41.65 |
| EAD (EN rule) | 0 | 82.46 | 44.35 | 82.95 | 42.92 |
| | 10 | 81.54 | 44.75 | 82.90 | 43.05 |
| | 20 | 76.37 | 31.95 | 83.66 | 43.31 |
| | 30 | 73.49 | 31.55 | 82.57 | 44.58 |
| | 40 | 75.29 | 36.89 | 83.39 | 43.05 |
| EAD ($L_1$ rule) | 0 | 82.41 | 44.85 | 82.67 | 42.67 |
| | 10 | 82.47 | 44.95 | 82.99 | 43.69 |
| | 20 | 75.68 | 31.14 | 82.55 | 43.56 |
| | 30 | 73.67 | 30.84 | 81.49 | 42.42 |
| | 40 | 76.04 | 37.19 | 82.10 | 42.80 |

