# OpenReview forum: "On the Limitation of Local Intrinsic Dimensionality for Characterizing the Subspaces of Adversarial Examples"
_ICLR.cc/2018/Workshop — Accept_

### Official Review · AnonReviewer3 · 2018-03-10
**The paper makes reasonable contribution in showing that LID may be limited in characterize the intrinsic dimension of the adversarial samples**

**Rating:** 6
**Confidence:** 2

**Review:**

This paper performs experimental study to show that LID proposed in a recent paper has significant limitations to identifying the intrinsic low dimensional space of adversarial samples. In particular, the authors show that LID is very sensitive to correctly specify the parameters, and moreover, when adversarial perturbations are generated via a different network, LID fails to find the low dimensional space.

---

### Official Review · AnonReviewer1 · 2018-03-11
**Experimental results showing limits of a recent measure of "attackability" of models**

**Rating:** 7
**Confidence:** 2

**Review:**

This paper showcases that Local Intrinsic Dimensionality (LID) is not always an adequate measure to characterize how a given model would respond to a given attack type. The experiments test the effectiveness of LID on MNIST and CIFAR-10, both 1) varying confidence thresholds, without knowing the attack parameters, and 2) transferring attacks from one model to another, knowing the attack parameters. The attacks used in this paper are Carlini and Wagner (C&W), and elastic-net (EAD).

It is unclear to me if LID was computed with an L1 distance (instead of L2) in the case of the EAD (L1 rule) attacks, or if it would help.

I do not know how exactly [Ma et al., 2018] was received in the adversarial examples community, but it will be presented as oral at this ICLR, so this workshop paper should yield an interesting discussion.

---

### Official Review · AnonReviewer2 · 2018-03-12
**Further adds to the analysis of LID**

**Rating:** 7
**Confidence:** 3

**Review:**

The authors conduct several experiments showing limitations of LID for characterizing adversarial attacks. The attacks are generated using different confidence levels. The contribution of the paper is not overwhelming, but adds to the understanding of LID. The authors take the most recent SOTA work by Athalye et al. into account and show that their main message still holds when using high-confidence adv examples at training time.

minor:
\citep and \citet are mixed up at several places

---

### Decision · Program_Chairs · 2018-03-20
**ICLR 2018 Workshop Acceptance Decision**

**Decision:**

Accept

**Comment:**

Congratulations, your paper was accepted to the ICLR workshop.